# Post-Emergence Water-Dispersal Application Provides Equal Herbicidal Activity against *Echinochloa crus-galli* and Rice Safety as Foliar Spraying of Penoxsulam

**DOI:** 10.3390/plants12234061

**Published:** 2023-12-03

**Authors:** Jinqiu Sun, Xiaoyue Yu, Hongxing Xu, Yongjie Yang, Mengjie Liu, Yanchao Zhang, Yongliang Lu, Wei Tang

**Affiliations:** 1State Key Laboratory of Rice Biology and Breeding, China National Rice Research Institute, Hangzhou 311400, China; 2Institute of Plant Protection and Microbiology, Zhejiang Academy of Agricultural Sciences, Hangzhou 310021, China; 3School of Information Science and Enzhegineering, Zhejiang Sci-Tech University, Hangzhou 310018, China

**Keywords:** penoxsulam, *Echinochloa crus-galli*, herbicide residue, ALS, GST, cytochrome P450

## Abstract

Penoxsulam is an acetolactate synthase (ALS)-inhibiting herbicide usually applied by post-emergence foliar spraying (PFS) for the control of *Echinochloa crus-galli* and numerous annual weeds in paddy fields. Herbicides applied by foliar spraying can have negative impacts on the environment, ecosystems, and human health. In this study, the response of *E. crus-galli* and rice to the PFS and post-emergence water-dispersal (PWD) applications of penoxsulam, and the differences in the detoxification displayed by them between the two treatment methods were compared. The results showed that the PWD application of penoxsulam provides a similar control efficacy against *E. crus-galli* as PFS at the 1-, 3-, and 5-leaf stages. Meanwhile, the PWD application had a higher safety for the rice. After being treated with 30 g a.i. ha^−1^ penoxsulam, residues were not detected in the rice treated by the PWD application method, whereas, with the PFS treatment, there was 59.0 µg/kg penoxsulam remaining. With the PFS application, there were many more residues of penoxsulam in the *E. crus-galli* than with the PWD method; the amount of residues was 32-fold higher 12 h after treatment. The in vitro enzyme activity assays revealed that the activities of ALS, glutathione-*S*-transferase (GST), and cytochrome P450 monooxygenases (P450) were increased in the PWD treatments, and were 1.5-, 1.3-, and 2.3-fold higher than with PFS 72 h after treatment. The real-time quantitative PCR (qRT-PCR) revealed that the *GST1* and P450 genes, *CYP81A14*, *CYP81A12*, *CYP81A18*, and *CYP81A21* were upregulated with the PWD application versus PFS in the *E. crus-galli*. In summary, these results demonstrate that the herbicidal activity was not affected by the upregulation of target and metabolic enzyme activities with the PWD application of penoxsulam. This research could contribute to application strategies reducing the risk of rice injury and environmental impacts by using water-dispersal formulations of penoxsulam.

## 1. Introduction

Herbicides are widely used in weed management because they provide better weed control, reduce labor costs, and do not alter the soil structure. However, the incorrect use of herbicides promotes the development of weed resistance and has a negative influence on environmental protection. Penoxsulam (2-(2,2-Difluoroethoxy)-N-(5,8-dimethoxy-[1,2,4] triazolo [1,5-c] pyrimidin-2-yl)-6-(trifluoromethyl) benzenesulfonamide) is a triazolopyrimidine sulfonamide (TSA) herbicide developed by Dow AgroSciences and applied in paddy fields. It inhibits the activity of acetolactate synthase (EC 4.1.3.18, ALS), which catalyzes the first reaction in the biosynthesis of branched-chain amino acids (isoleucine, leucine, and valine), and provides excellent herbicidal activity against annual grass weeds such as *Echinochloa crus-galli*, broadleaf weeds, and sedges in rice [1]. As of January 2022, a total of 337 penoxsulam products (including technical and formulations) have been registered in China according to the China Pesticide Registration Watch (CPRW). It is used mainly as a post-emergence foliar spray formulation. However, aerial spraying directly applies herbicides on crop seedlings, which may cause crop seedling injury. Previous studies have shown that the foliar application of penoxsulam negatively affects the roots and leaves of the seedlings of some rice varieties [2]. On the other hand, herbicide aerial spraying may result in the drift to neighboring areas, thus possessing a high risk of drift injury and environmental contamination [3]. It is also reported that sub-lethal doses of herbicides from spray drift may lead to the resistance evolution of weeds [4], through which *Lolium rigidum* [5,6,7], *Raphanus raphanistrum* [8], and *Amaranthus palmeri* [9] have evolved herbicide resistance under the selective pressure of low-dose herbicides over a long period of time. In addition, sublethal doses of herbicides affect non-target flowering plants [10].

Compared with foliar spraying, water-dispersal formulations, such as “1 kg granule”, “diffusion granule”, “Jumbo”, and “Flowable” have been developed to avoid drift problems and save farmers the trouble of applying herbicides [11]. In China, there are 19 granular products registered among 238 formulations of penoxsulam. The entry route is different for herbicides applied by water-dispersal and foliar spraying. The leaf absorption of herbicide after spraying requires the active ingredient to be retained in the leaf and remain there for a sufficient time until it is absorbed. This process was influenced by factors such as the spray equipment [12], spray drops characteristics [13,14], and plant leaf surface characteristics [15,16,17]. When the herbicide is applied using post-emergence water-dispersal formulations, the absorption of the herbicide may occur through the stem after being dissolved in the water, and the active ingredient can then be translocated to the apical meristem.

With the wide use of mechanical rice transplanters, the timing of rice seedling transplanting has shifted from the 4- to 6-leaf stage (grown seedlings) to the 2- to 3-leaf stage (young seedlings). As a result, an extended period of time is needed for the creation of shade by the standing rice seedlings, causing the prolonged requirement of weed control [11]. *E. crus-galli* is one of the most prevalent and troublesome weed species in rice production in China, which can cause as much as an 80% rice yield reduction if uncontrolled [18,19]. Since *E. crus-galli* has morphologic similarities to rice, and along with its extended emergence throughout the growing season, high photosynthetic capacity, and prolific seed production, this makes it highly competitive and hard to control in paddy fields [20]. Penoxsulam can be absorbed through both the leaf and stem of plant seedlings; the herbicidal activity and rice safety between the different application methods of penoxsulam have yet to be identified.

To achieve better weed control efficacy, a specific field environment is required to coordinate with the herbicide application [21,22,23]; for instance, the field needs to maintain a certain water depth when applied with water-dispersal formulated herbicides, and be kept drained for foliar spraying herbicides. When water layers are kept in the field, weeds have to adapt their metabolism in order to avoid energy shortage. The difference in herbicide metabolism has been identified and the genes responsible have been cloned in *Echinochloa* species such as *CYP81A68* [24], *CYP81A12*, and *CYP81A21* [25]. When treated with the same dose of herbicide using different application methods, in conditions where water is involved, insight into the different herbicide responses in *E. crus-galli* is lacking.

Previous studies have mostly concentrated on the weed control efficacy evaluation of different penoxsulam formulations or combinations [26], or have either focused on the resistance level and molecular mechanism of penoxsulam [27,28,29] in *Echinochloa* species, or the environmental effects of penoxsulam [30]. Understanding how penoxsulam is absorbed and translocated in *E. crus-galli* and rice plants after being treated with different application methods, and how penoxsulam influences the target and key metabolic enzymes and their genes in *E. crus-galli*, is a key step toward the effective post-emergence application of penoxsulam using water-dispersal formulations. In this study, rice and *E. crus-galli* seedlings at the same growth stage were divided into two groups, kept in moist soil or 2 cm deep water, and then treated with penoxsulam either by foliar spraying or water-dispersal, respectively. The objectives of this study were to (a) investigate the response to penoxsulam in rice and *E. crus-galli* seedlings; (b) determine the variation in the absorption and metabolism of penoxsulam in *E. crus-galli* and rice; (c) determine the differences in the enzyme activity of ALS, GST, and P450, and the expression of the *ALS*, *GST,* and *P450*s genes in *E. crus-galli* using qRT-PCR under the two application methods of penoxsulam.

## 2. Results

### 2.1. Dose–Response Experiments to Penoxsulam by PFS and PWD Application

The effect of the penoxsulam in *E. crus-galli* at the 1- to 5-leaf stage applied using two application methods was evaluated. Obvious yellowing was observed on the young leaves of the *E. crus-galli* 5 d after treatment (DAT) with PFS, followed by stunted growth and wilting at 10 DAT. Similar symptoms were observed 2 days later with the PWD treatments. The results of the whole-plant dose–response assay indicated that the PWD application was as effective as the PFS application in controlling the *E. crus-galli* at the 1-, 3-, and 5-leaf stages (Figure 1). As shown in Figure 2a, penoxsulam at the recommended field dose of 30 g a.i. ha^−1^ controlled 100% of the *E. crus-galli* at the 1-leaf stage after the two application methods. Penoxsulam with the PWD and PSF application at 7.5, 15, and 30 g a.i. ha^−1^ caused a more than 90% fresh weight reduction in the *E. crus-galli* at the 3-leaf stage (Figure 2b). The control efficacy of the PFS and PWD application methods, as determined by an ED_50_s value of 1.3 and 1.4, respectively, for the *E. crus-galli* at the 5-leaf stage, is illustrated in Figure 2c.

### 2.2. Effects of Two Application Methods on Growth of Rice

After 21 days of the penoxsulam treatment, the above-ground fresh weight of the rice was assessed to evaluate the safety of the two application methods on the rice. There was found visible damage to the rice at the seedling stage growth. The inhibition rate of the herbicide on the above-ground fresh weight of the rice gradually decreased with the growth of the rice seedlings. At the 1-leaf stage, the application of 30 g a.i. ha^−1^ penoxsulam through the PFS and PWD methods resulted in an inhibition rate of 50.4% and 42.7%, respectively, while, at the 5-leaf stage, it reached 31.6% and 27.6%. Furthermore, at 15 and 30 g a.i. ha^−1^, the fresh weight inhibition rate using the PWD method consistently remained lower than that using the PFS method at all three leaf stages (Table 1), indicating a higher safety of the PWD method compared with the PFS method at the recommended field doses.

### 2.3. Differences in Residues of Penoxsulam in Rice and E. Crus-Galli after PFS and PWD Treatments

The absorption of penoxsulam in the rice and *E. crus-galli* was evaluated by assaying the residues at 12, 24, 72, and 120 h after treatment. No residues of penoxsulam were detected in the rice treated with the PWD method at doses of 7.5 and 30 g a.i. ha^−1^. However, even at the low dose (7.5 g a.i. ha^−1^), 17.0 ug/kg penoxsulam residues could be detected in the rice after treatment with the PFS method at 12 h post-treatment. After treatment with the PFS method, the residue levels peaked in the *E. crucis-galli* after only 12 h, before decreasing over time. However, the residue levels remained significantly higher than those observed following treatment with the PWD method in both the rice and *E. crucis-galli* seedlings. The level of residual penoxsulam found within the *E. crucis-galli* was approximately 5.4-fold greater than that found within the rice when using the PFS method 12 h after treatment (Table 2).

### 2.4. Acetolactate Synthase (ALS), Cytochrome P450s (P450), and Glutathione S-Transferases (GST) Activity Assay

The results revealed that, in the absence of herbicide treatment, there was no significant disparity in the ALS and GST activity between the PFS and PWD methods; however, the P450 activity of the PWD method exhibited higher levels compared to that of the PFS method. In the *E. crus-galli*, after treatment with the PFS method, the peak activities of ALS, P450, and GST were observed at 24 h. Conversely, for the PWD method, the ALS and P450 enzyme activities reached their peak at 72 h, while the GST activity continued to increase over time. Notably, except for 24 h post-treatment, all three enzymes’ activities were higher following the PWD treatment than those resulting from the application of the PFS method (Figure 3). These results suggested a potential association between the differences in the peak ALS activity and the variations in the symptom presentation timing caused by the different methods employed. Moreover, the elevated levels of both the P450 and GST activity may be attributed to their involvement in herbicide metabolism.

### 2.5. Expression Analysis of ALS Gene and Metabolism-Related Genes

The trend of ALS gene expression following the penoxsulam treatment was consistent between both methods, with the expression levels being upregulated and reaching a peak at 12 h post-treatment. Specifically, the expression levels were 27.0 and 27.7 times higher than those of the control in the PFS and PWD methods, respectively (Figure 4a). Moreover, the magnitude of the change in gene expression was greater for the PFS method compared to the PWD method. The expression levels of the *GST1* and *CYP81A14* genes exhibited an upward trend at 12 h after treatment, with the PFS method reaching its peak at 24 h and the PWD method peaking at 72 h. Notably, the gene expression levels were higher in the PWD method compared to the PFS method. However, there was no significant difference observed in the *GST1* gene expression between these two methods (Figure 4b). In contrast, the *CYP81A14* gene expression level in the *E. crus-galli* was significantly higher after the treatment with the PWD method compared to the PFS method at both 24 and 72 h post-treatment (Figure 4c). Furthermore, metabolism-related genes (*CYP81A12*, *CYP81A18*, and *CYP81A21*) also exhibited higher expression levels 24 h after treatment using the PWD method compared to the PFS method (Figure 4d).

## 3. Discussion

*E. Crus galli* is one of the most troublesome weeds in rice fields around the world due to its strong competitiveness, high reproductive capacity, and wide ecological adaptability [18]. The post-emergence application method for herbicides adopted by most farmers for the control of *E. crus-galli* in China is spraying. After the deposition of the droplet containing penoxsulam on the plant leaves, the surface characteristics, shape, and orientation of the target leaves are considered important factors for the retention and amount of herbicide absorption. The cuticle layer, apoplast, cell wall, and membrane are barriers on the way to the meristematic tissue, where ALS activity is inhibited by penoxsulam. Compared with water-dispersal application, in which herbicides are absorbed by the stem of young plants, foliar spraying has a larger contact area and shorter translocation distance. It is hypothesized that the foliar spraying of penoxsulam may show a higher control efficacy on *E. crus-galli* than water dispersal. The results of this study showed that there is no significant difference between the efficacy of the water-dispersal and foliar spraying of penoxsulam. It is probable that the young seedling stem has a less developed cuticle, devoid of a wax layer, making plant more permeable to penoxsulam [31]. Unexpectedly, but not surprisingly, it was noted that, after 21 days of treatment, foliar spraying caused a higher above-ground fresh weight reduction in the rice seedlings at all three treated leaf ages than the water-dispersal treatments. 

In the current study, much less remaining penoxsulam was detected in both the rice and *E. crus-galli* seedlings treated by water-dispersal than foliar spraying at the same dose. When herbicides are applied to water, the dissolved active ingredients and active ingredients bound to soil particles can accumulate in rice field sediments. Penoxsulam is a non-volatile compound; the adsorption reached equilibrium after 4 h in paddy soil, and the adsorption coefficients were related to the soil pH [32,33]. Moreover, the dissipation time of penoxsulam in water is relatively short; 45~55% of the initial dose dissipates 6 h after treatment [34]. It is also reported that only 12.8% and 10% of the penoxsulam is detected 2 days after 0.07 mg/kg was mixed with soil [35]. It is clear that a sufficient lethal dose could have been absorbed and translocated to the meristematic tissues of the *E. crus-galli* when the penoxsulam was applied by water-dispersal methods. 

Our results indicated that, under foliar spray application, the uptake of the penoxsulam was much higher in the *E. crus-galli* than the rice. This may be due to the wax content on the surface of the *E. crus-galli* and rice leaves. Previous studies have pointed out that the area covered by pesticides on the plant surface is negatively correlated with the wax content [36], and that the wax content of *E. crus-galli* leaves is much higher than that of rice [37]. These results suggested that the water-dispersal application of penoxsulam has a minor impact on the water and soil environment, or the safety for subsequent crops. Therefore, it is clear that post-emergence water-dispersal application methods or formulations are practical strategies for the application of penoxsulam in rice.

Plants have a strong metabolic detoxification capacity for ALS-inhibiting herbicides, and this rapid detoxification metabolism may be possessed by the plant itself or induced subsequently [38]. After herbicide treatment, the ALS and GST activities and their related gene expression levels in the *E. crus-galli* showed a tendency to increase and then decrease, which was observed in this study. The increase in the metabolic enzyme activity was similar to the findings that the enzymes ALS [28] and GST [39] are involved in the increased metabolic detoxification after treatment with ALS inhibitors in *E. crus-galli*. Compared with the PFS treatment, the activities of ALS and GST reached their peak later than with the PWD treatment; this was inconsistent with the symptoms of obvious herbicidal injury being observed 2 days later with the PWD treatments. The relative higher activity of GST in the PWD treatment may due to the flooding conditions during the experiment, which caused a GST-involved stress response in the *E. crus-galli*. Further, we found that the *ALS* and *GST1* genes were upregulated in the PWD treatment relative to the PFS treatment. Previous studies have shown that the upregulation of the *GST2* gene in *E. crus-galli* [40], the *GST* genes in *Alopecurus aequalis* [41], and the *GSTDHAR2* and *MGST1* genes in *Malachium aquaticum* [42] was identified to be associated with herbicide metabolic resistance. The direct evidence that the PWD treatment with penoxsulam triggered higher GST and ALS activities than the PFS treatment needs further study. 

Plant P450s constitute one of the largest families of protein genes involved in metabolizing varied biotic and abiotic xenobiotics including herbicides, decreasing their phytotoxocity [43]. A considerable number of CYP450 genes have been reported to be involved in non-target site resistance to varied herbicides in different weed species. For instance, CYP76C1, CYP76C2, and CYP76C4 can metabolize phenylurea herbicides [44], and CYP81A12 and CYP81A21 can metabolize penoxsulam and bensulfuron [45]. The CYP81 family has only been reported in grass species. Dimmano et al. [46] reported that CYP81A12, CYP81A14, and CYP81A21 could metabolize penoxsulam via *O*-demethylated in *E. phyllopogon*. Similarly, the over-expression of *CYP81A68* in *E. crus-galli*, was identified to confer metabolic resistance to penoxsulam [24]. Our present work provided the first evidence that the expression of *CYP81A14*, *CYP81A12*, *CYP81A18*, and *CYP81A21* were higher after the water-dispersal treatment than after the foliar spraying of penoxsulam in *E. crus-galli* seedlings. The higher expression of the CYP81 genes may contribute to the rapid metabolism of penoxsulam, which could be the result of the relatively lower herbicidal activity against *E. crus-galli* in the water-dispersal treatments.

The relationship between flooding stress and the expression of the CYP81 genes was not examined in this current study. We hypothesized that it might be that the flooding conditions, in which the water layers were kept during the water-dispersal treatments, induced higher enzyme activity and gene expression levels. There are few studies reporting on the relationship of flooding stress and the activities of ALS, GST, P450s and the related gene expression levels in weeds. However, previous studies have found that the GSTase-related genes in alfalfa [47] and cucumber [48], and the genes encoding the P450 proteins are upregulated under waterlogging stress. As the *E. crus-galli* could be totally controlled with 30 g a.i./ha of penoxsulam before the 5-leaf stage, and there were no significant differences between the ED_50_s of the PWD and PFS methods, this may justify its relatively lower herbicidal activity at the same dose in the PWD method. 

## 4. Materials and Methods

### 4.1. Plant Materials and Growth Conditions

Seeds of *E. crus-galli* were collected in a rice field in Baimi town, Taizhou city, Jiangsu province, China (E120.258, N32.514), in 2019. Rice seeds (variety XIU SHUI 134) were provided by the Seed Bank of the China National Rice Research Institute (CNRRI). The seeds used for the experiments were disinfected with 3% sodium hypochlorite for 5 min, and then washed three times with distilled water. The seeds were placed in 9 cm diameter Petri dishes containing two layers of filter paper (Double Ring #102, Hangzhou Special Paper Industry Co., Ltd., Hangzhou, China), and moistened by 4 mL of distilled water. The Petri dishes were transferred into a controlled environment growth incubator (GXZ-300C, Dongnan Instrument Manufacture, Ningbo, Zhejiang, China) at day and night temperatures of 30/20 °C with 12 h light. The photoperiod was set to coincide with the high temperature period. A total of 10 uniform germinating seeds (with 2 mm emerged radicles) were sown on the surface in plastic pots (13.5 cm diameter, 11.0 cm height) containing potting soil (Hangzhou Jin Hai Agriculture Co., Ltd., Hangzhou, China). After sowing, the seeds were covered with a thin layer of soil and moisture was maintained by irrigating with an overhead sprinkler every day. After seedlings emerged, seedlings were thinned to three plants in each pot, with three replicates for each treatment. These pots were placed in a greenhouse at CNRRI (E119.935, N30.078), where the temperature was kept at 25/15 °C, and the humidity was higher than 50% under natural light. 

When the plants grew to 1-, 3- and 5-leaf stage, seedlings were treated with the commercial formulation of penoxsulam (2.5% OD, Corteva Agriscience, Shanghai, China) by two methods: (1) post-emergence foliar spraying (PFS): seedlings were sprayed using a laboratory sprayer equipped with a flat-fan nozzle (TP6501E) to deliver 200 L ha^−1^ at 230 kPa; (2) post-emergence water-dispersal (PWD): penoxsulam was diluted and added using a pipette into each pot kept in 2 cm deep water. 

### 4.2. Dose–Response Experiments to Penoxsulam by PFS and PWD Application

To investigate the response of *E. crus-galli* and rice to penoxsulam at different growth stages under the two application methods mentioned above, whole-plant dose–response experiments were conducted in the greenhouse. The treatment doses of penoxsulam were 0, 0.24, 0.48, 0.96, 1.88, 3.75, 15, and 30 g a.i. ha^−1^ for rice and *E. crus-galli* seedlings at 1- and 3-leaf stages, and 0, 0.94, 1.88, 3.75, 7.5, 15, 30, 60, 120 g a.i. ha^−1^ for seedlings at 5-leaf stage. The soil was maintained as wet for PFS treatments and kept in 2 cm deep water for PWD treatments throughout the experiments. After 3 weeks of treatment, the number of surviving seedlings and fresh weight of above-ground material of each *E. crus-galli* and rice plant was harvested, and fresh weight was recorded.

### 4.3. Penoxsulam Absorption in E. crus-galli Using HPLC-Q-TOF-MS

When plants reached 3-leaf stage, they were treated with the two application methods at 7.5 g a.i. ha^−1^ of penoxsulam. The above-ground tissues were harvested at 12, 24, 72, 120 h after treatment. Treated plants were washed three times with distilled water to remove unabsorbed penoxsulam. The water on the surface of the plants was blotted out with filter paper, placed in liquid nitrogen for quick freezing, and then stored at −80 °C until use. A total of 5 g of above-ground tissue was used for each replicate and three replicates were used for analysis. Frozen plant material was ground into a powder using a mortar and pestle in liquid nitrogen. The extraction and isolation of penoxsulam was performed as described previously [49]. Both treatment groups were analyzed using ultra performance liquid chromatography mass spectrometry (UPLC-MS/MS) according to Pan et al. [24] and Feng et al. [39].

### 4.4. In Vitro Assay of ALS, GST, and Cytochrome P450 Activities in E. crus-galli

To determine the effects of the two application methods on the activities of target enzyme ALS, or key metabolic enzyme GST and P450 of *E. crus-galli*, seedlings were cultivated to 3-leaf stage as described above. Seedlings were treated with penoxsulam at 7.5 g a.i. ha^−1^ and 2 g fresh leaf tissue was collected at 0, 12, 24, 72, and 120 h after treatment. The leaf tissue was treated with PBS prior to biochemical assays after being ground with liquid nitrogen. A fresh leaf sample (0.1 g) was homogenized with 0.9 mL of PBS [100 mmol/L Tris-HCl, (pH 8.3), 300 mmol/L glycerinum, 5 mmol/L DTT, 2 mmol/L EDTA, 0.5 mmol/L PMSF, and 0.01% (*v/v*) Triton X-100] at pH 7.2–7.4 and centrifuged at 3500 rpm for 15 min at 4 °C. The supernatant was collected in a centrifuge tube and placed in an ice bath. Activities of ALS, GST, and P450 were determined by using enzyme-linked immunosorbent assay (ELISA) kits (Meimian Biotechnology Co., Ltd., Yancheng, China) according to the manufacturer’s instructions. Each treatment included three replications, and the experiment was repeated once.

### 4.5. ALS, GST, and Cytochrome P450 Gene Expression Analysis in E. crus-galli

The above-ground tissues of plants treated with the two application methods at the 3- or 4-leaf stage were harvested at 0, 12, 24, 72, and 120 h after treatment. Total RNA was extracted by using the RNAprp Pure Plant Extraction Kit (Tiangen, Beijing, China). The cDNA synthesis kit (TAKARA, Beijing China) was then used to reverse transcribe the RNA samples. qRT-PCR was performed with an iTaq Universal SYBR Green Supermix (Bio-Rad, Hercules, CA, USA) on the QuantStudioTM 1 real-time PCR System and with the following thermal cycle conditions: denaturing at 95 ^◦^C for 120 s followed by 40 cycles of 95 ^◦^C for 15 s and 60 ^◦^C for 15 s. *ALS* [50], *GST1*, *CYP81A12*, *CYP81A14*, *CYP81A18,* and *CYP81A21* [39] were amplified using published primers (Table 3). There were three replicates for each treatment, and the relative expression was calculated using the 2^(−∆∆C(T))^method [51].

### 4.6. Statistical Analysis

ED_50_ values (50% plant mortality due to herbicides) were estimated by nonlinear regression analysis using origin software (Version 2018). In this experiment, the response curve of weed above-ground biomass to the penoxsulam dose was fitted to a four-parameter logistic model (1) [52].
(1)$$y=C+\frac{D-C}{{1+(x/{ED}_{50})}^{b}}$$
here, *C* refers to the lower limit of plant survival at the maximum herbicide dose, and *D* is the upper limit of plant survival at doses close to the control treatment. *x* is the dose of the herbicide, *b* represents the relative slope of *x* (the dose that reduces the above-ground fresh weight by 50%). *y* is the percentage of fresh weight of plants in the treatment group to the fresh weight of plants in the control group. After 21 days of treatment, the above-ground fresh weight of rice was measured and the fresh weight inhibition rate was calculated according to Equation (2).
(2)$$E/\%=\frac{W1-W2}{W1}\times 100$$
here, *E* is the fresh weight inhibition rate, %; *W*1 is the fresh weight of rice in control group (g); *W*2 is the fresh weight of rice in treatment group (g). Significant differences of inhibition rate of rice fresh weight, ALS, P450 and GST enzyme activity and gene expression data were determined by the *t*-test (α = 0.05). Significant differences of remaining penoxsulam data were determined by one-way ANOVA, Duncan test (*p* < 0.05).

## 5. Conclusions

In summary, compared to foliar spraying, penoxsulam showed a similar efficacy against the *E. crus-galli* via water-dispersal application. Furthermore, as a result of less penoxsulam remaining in the rice seedlings, the post-emergence water-dispersal application method might be safer than the foliar spraying of penoxsulam in rice. The enzyme activities of ALS, GST, CYP450, as well as their related gene expression levels were up-regulated in the PWD treatments; there were no significant differences between the ED_50_s of the PWD and PFS applications of penoxsulam. In China, the environmental pollution and human health problems caused by herbicide spraying have attracted public attention, and understanding the efficacy of penoxsulam via water dispersal is necessary to explore its application scope. As such, the environmental impacts of penoxsulam spraying could be reduced by using non-spraying tactics.

## Figures and Tables

**Figure 1 plants-12-04061-f001:**
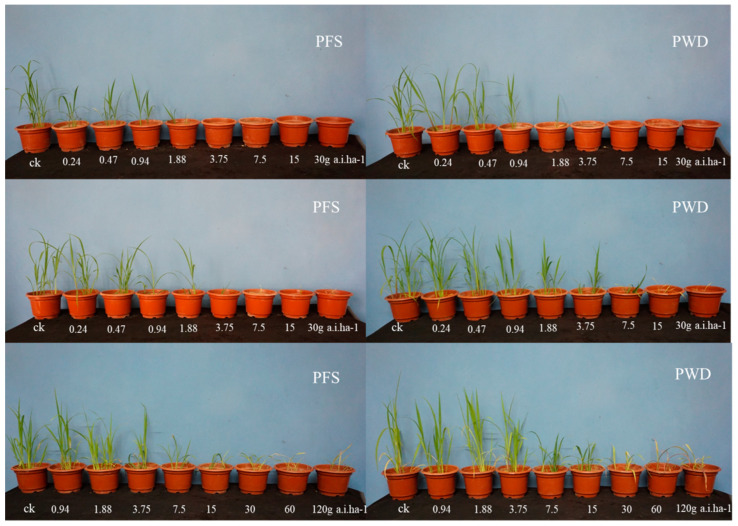
Photographs of barnyard grass after 21 days of treatment with different doses of penoxsulam by post-emergence foliar spraying (PFS) and water-dispersal application (PWD). The first, second, and third rows represent *E. crus-galli* at 1- (first row), 3- (second row), and 5-leaf stages (third row), respectively.

**Figure 2 plants-12-04061-f002:**
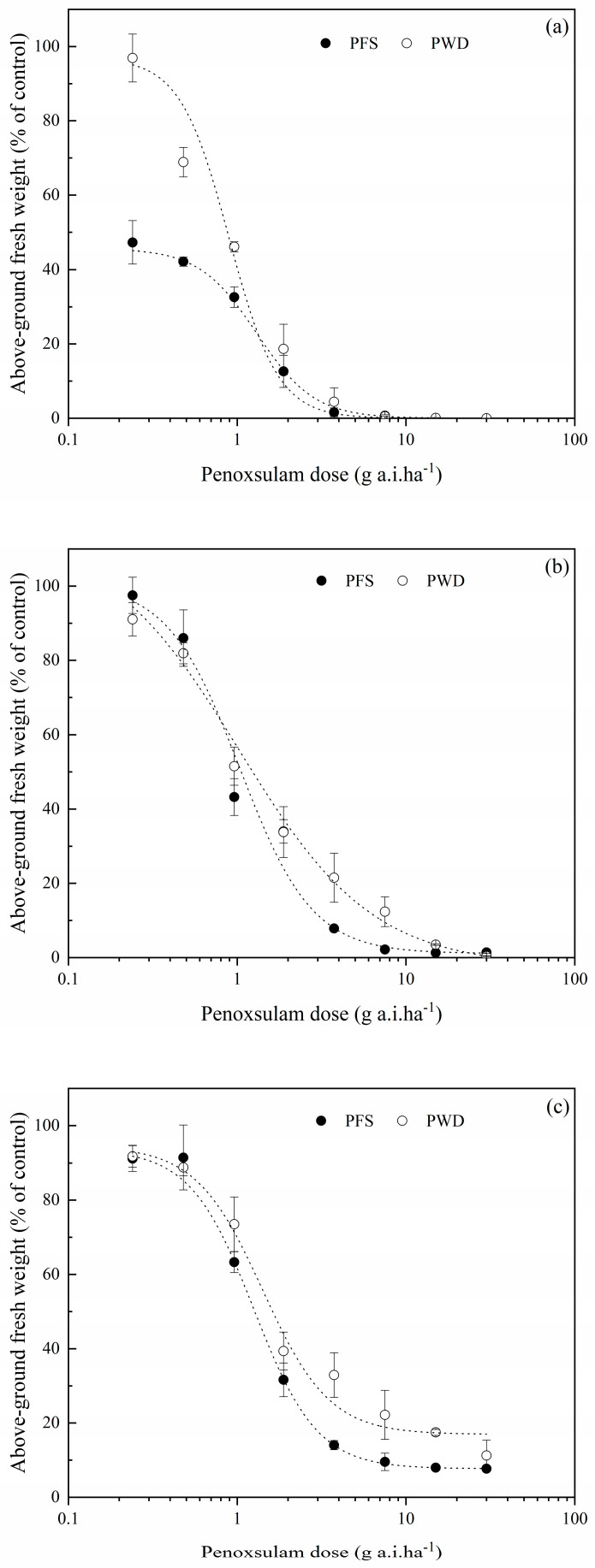
Dose–response curve of *E. crus-galli* at 1-leaf stage (**a**), 3-leaf stage (**b**) and 5-leaf stage (**c**) at different doses of penoxsulam by post-emergence foliar spraying (PFS) and water-dispersal application (PWD). Each point represents the mean ± SE of twice-repeated experiments, each including three replicates.

**Figure 3 plants-12-04061-f003:**
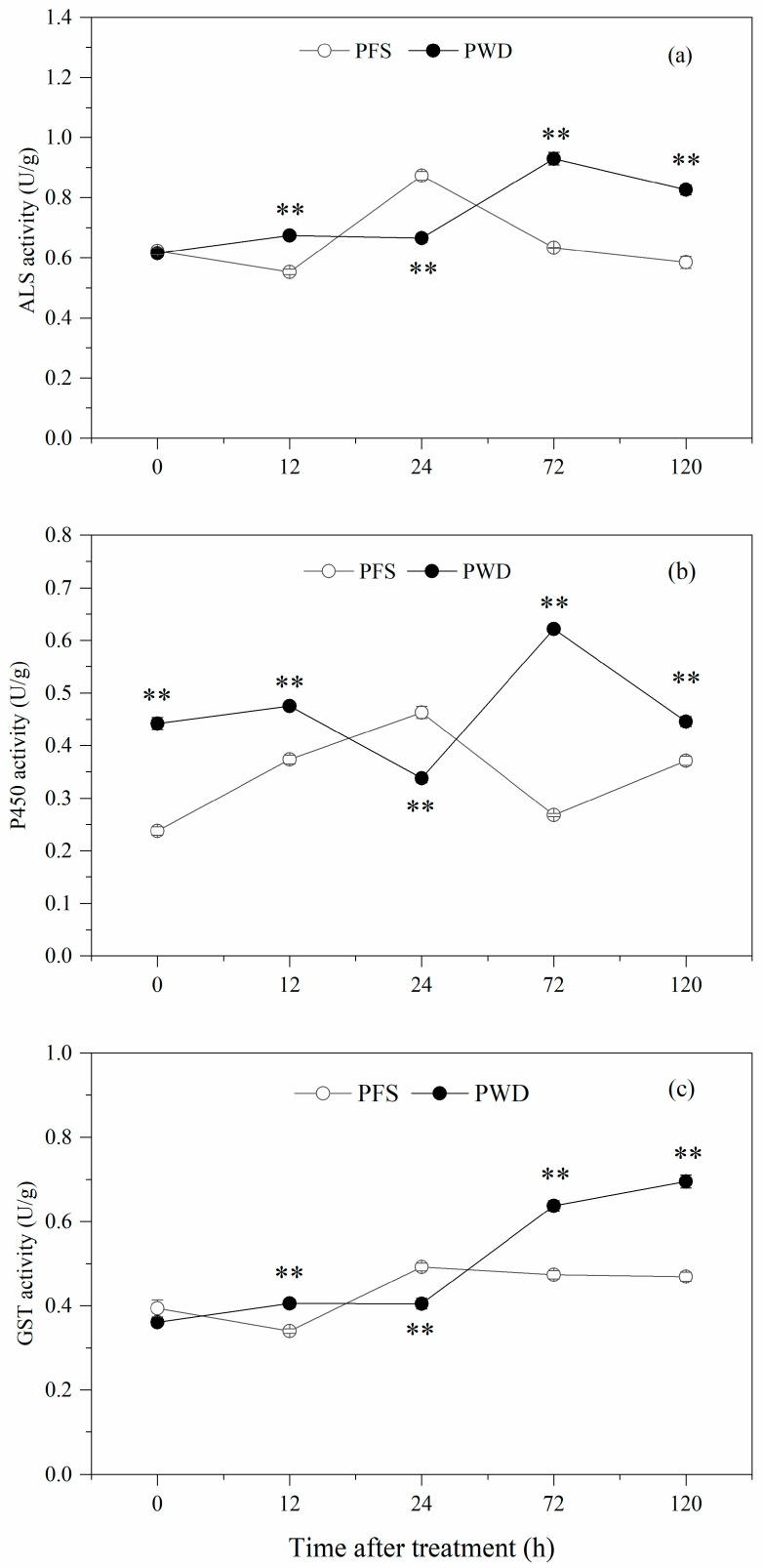
Activity of ALS (**a**), P450 (**b**), and GST (**c**) at 0, 12, 24, 72, and 120 h after treatment with penoxsulam by post-emergence foliar spraying (PFS) and water-dispersal application (PWD). Each point refers to mean ± SE of twice-repeated experiments, each including three replicates. The significance between the two methods was detected by *t*-tests (** *p* < 0.01).

**Figure 4 plants-12-04061-f004:**
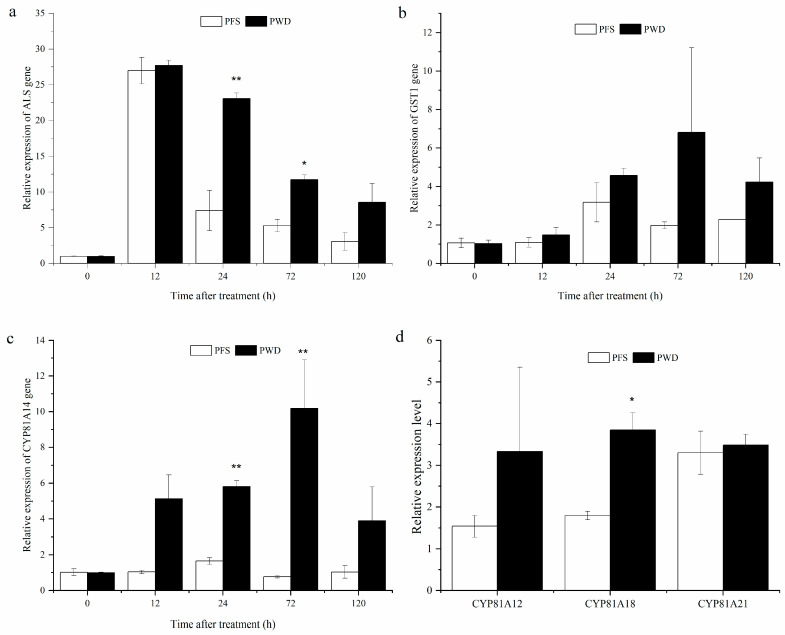
The gene expression levels of *ALS* (**a**), *GST1* (**b**), and *CYP81A14* (**c**) at different time intervals, and the relative gene expression levels of *CYP81A12*, *CYP81A18,* and *CYP81A21* after 24 h treatment by post-emergence foliar spraying (PFS) and water-dispersal application (PWD) of penoxsulam treatment (**d**). Data are presented as mean ± SE; significant differences between the two methods were detected by *t*-tests (* *p* < 0.05, ** *p* < 0.01).

**Table 1 plants-12-04061-t001:** Fresh weight inhibition rate in rice at recommended field doses.

Treat Target	Dose (g a.i.ha^−1^)	Inhibition Rate of Rice Fresh Weight ± SE(%)		*p*-Value
		PFS	PWD	
1-leaf-stage rice	15	38.3 ± 7.3	28.0 ± 6.5	0.299
	30	50.4 ± 0.9	42.7 ± 7.0	0.313
3-leaf-stage rice	15	31.5 ± 9.0	23.5 ± 6.0	0.424
	30	47.0 ± 2.7	36.5 ± 3.2	0.024
5-leaf-stage rice	15	29.0 ± 5.6	17.1 ± 4.6	0.150
	30	31.6 ± 4.9	27.6 ± 1.3	0.454

Data refer to mean ± SE and values were the means of three replicates. PFS and PWD stand for post-emergence foliar spraying and water-dispersal application of penoxsulam, respectively. *p*-value for the inhibition rate of the two application methods at the same dose was determined by *t*-tests.

**Table 2 plants-12-04061-t002:** Remaining penoxsulam in barnyard grass and rice was determined by HPLC- MS/MS analysis at 12, 24, 72, and 120 h after treatment.

Target	Dose (g a.i.ha^−1^)	Method	Remaining Penoxsulam (µg/kg)			
			12 h	24 h	72 h	120 h
Rice	7.5	PFS	17.0 ± 5.7 a	4.0 ± 5.7 b	4.0 ± 5.7 b	0.0 ± 0.0 b
		PWD	ND	ND	ND	ND
	30	PFS	59.0 ± 5.7 a	39.5 ± 2.5 b	39.0 ± 6.0 b	20.5 ± 3.5 c
		PWD	ND	ND	ND	ND
*E.crus-galli*	7.5	PFS	21.5 ± 2.5 a	20.0 ± 0.6 a	13.5 ± 1.5 b	0.0 ± 0.0 c
		PWD	ND	ND	ND	ND
	30	PFS	320.0 ± 20.0 a	76.0 ± 9.1 b	34.0 ± 4.2 c	29.7 ± 4.4 c
		PWD	10.3 ± 0.3 c	11.7 ± 0.9 c	16.5 ± 1.5 b	20.3 ± 1.2 a

ND means penoxsulam was not detected by UPLC- MS/MS. The data represent the means ± SE. Different letters indicate significant differences in different time groups according to one-way ANOVA, Duncan’s test, *p* < 0.05.

**Table 3 plants-12-04061-t003:** PCR primers for qPCR gene expression level analysis.

Primer	Sequence (5′–3′)	References
β-actin-F	TTGCCTACATTGCCCTTGACTA	(Iwakami et al. 2014 [25])
β-actin-S	GAACCACCACTGAGGACGACA	
ALS-F	TGGGGCTATGGGATTTGGTT	GenBank, KY071206.1
ALS-R	GCACAAAGACCTTCACTGGG	
GST1-F	AACGCAATGGCAGGTCTGAA	GenBank, JX122857.1
GST1-R	TACCGTTGTGGATGAGCACG	
CYP81A12-F	CACCCGGAGAAGCTCAAAAG	GenBank, AB818461
CYP81A12-R	ATGATGCTCTGGAGGTAGCC	
CYP81A14-F	AAGAACGACCTCCCCCATCT	GenBank, AB733994
CYP81A14-R	GGATGGCATACGCATTGACG	
CYP81A18-F	ATGCCATTCGGGATGGGAAG	GenBank, AB733996
CYP81A18-R	AGAGCTTCCAAAGGGACGAC	
CYP81A21-F	CAACCTGTGGGACTACCTGC	GenBank, AB818462
CYP81A21-R	GCACGGCAATCATGCTCTTC	

## Data Availability

The data will be made available on request.

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
