# Peer review of "Post-Emergence Water-Dispersal Application Provides Equal Herbicidal Activity against Echinochloa crus-galli and Rice Safety as Foliar Spraying of Penoxsulam"

_plants, 2023, doi:10.3390/plants12234061_

Round 1
Reviewer 1 Report
Comments and Suggestions for Authors
Paper needs technical corrections:
Line 2: water-dispersal
Line 102: penoxsulam
Figure 2 a, b, c: Above-ground fresh……
Line 120: E. crus-galli – italic
Line 125: above-ground
Line 128: above-ground
Line 150: 12 h (Table 3).
Line 152: 72 and 120 h after treatment.
Table 3: Dose (g.ai.ha-1)
Line 162: 24 h
Line 163: 72 h
Line 164: 24 h
Figure 3 (a), (b), (c): activity (U/g)
Line 178: (Fig. 4a).
Line 184: (Fig. 4b).
Line 186: (fig. 4c).
Line 189: (Fig. 4d).
Figure 4 a, b: Time after treatment (h)
Line 195: t-tests (*P<0.05,**P<0.01)
Line 197: E. crus galli
Line 205: water-dispersal
Line 206 and 207: check the whole sentence?
Line 208: water-dispersal
Line 212: above-ground
Line 220: 6 h
Line 232: water-dispersal
Line 257: penoxsulam and
Line 294: The pots were not are
Line 295: the temperature was not is and humidity was not is …
Line 315: The above-ground
Line 339: The above-ground
Line 354: above-ground
Line 362: above-ground
Line 379: investigation
Line 380: formal analysis and investigation
Line 281: funding
Line 382: investigation
Line 430: (Echinochloa crus-galli)
Line 434: barnyardgrass (….) populations
Line 448: penoxsulam
Line 452: Eisenia fetida – italic
Line 454: position 18 and 28 this same
Line 494: empty space
Comments on the Quality of English LanguageWell written paper.
Author Response
|
Thank you for your letter and for the reviewers’ comments concerning our manuscript entitled "Post-emergence water dispersal application provides equal herbicidal activity to Echinochloa crus-galli and rice safety as foliar spraying in penoxsulam". (ID: plants-2708058). Those comments are all valuable and very helpful for revising and improving our paper, as well as the important guiding significance to our researches. We have studied comments carefully and have made corrections which we hope meet with approval. Revised portion are marked in colored words or using the track changes mode in the paper. The main corrections in the paper and the responds to the reviewer’s comments are as flowing: Responds to the reviewer’s comments: Point-by-point response to Comments and Suggestions for Authors |
|
Comments 1: Line 2: water-dispersal, Line 205: water-dispersal, Line 208: water-dispersal, Line 232: water-dispersal |
|
Response 1: Thank you for pointing this out. We agree with this comment. Therefore, we have modified the water dispersal located on Line2, Line205, Line208 and Line232 to water-dispersal.
|
|
Comments 2: Line 102: penoxsulam, Line 257: penoxsulam and, Line 448: penoxsulam |
|
Response 2: Agree. We have corrected "penoxsulam" in the correct spelling and format
Comments 3: Figure 2 a, b, c: Above-ground fresh……, Line 125: above-ground, Line 128: above-ground, Line 212: above-ground, Line 315: The above-ground, Line 339: The above-ground, Line 354: above-ground,Line 362: above-ground Response 3: Agree. We have changed aboveground to above-ground.
Comments 4: Line 120: E. crus-galli – italic, Line 197: E. crus galli, Line 430: (Echinochloa crus-galli), Line 452: Eisenia fetida – italic Response 4: The Latin species name is formatted as required, but Eisenia fetida in Line452 is not modified as Eisenia Fetida-italic, because the reference title is Eisenia fetida.
Comments 5: Line 150: 12 h (Table 3), Line 152: 72 and 120 h after treatment, Line 162: 24 h ,Line 163: 72 h, Line 164: 24 h, Line 220: 6 h Response 5: Agree. According to the reviewer's opinion, we have rewritten in the correct format.
Comments 6: Table 3: Dose (g.ai.ha-1), Line 195: t-tests (*P<0.05,**P<0.01), Line 434: barnyardgrass (….) populations, Figure 3 (a), (b), (c): activity (U/g), Figure 4 a, b: Time after treatment (h) Response6: Agree. We have modified the correct format of the text and parentheses
Comments 7: Line 178: (Fig. 4a)., Line 184: (Fig. 4b).,Line 186: (fig. 4c).,Line 189: (Fig. 4d). Response 7: The content in parentheses has been modified as requested
Comments8: Line 206 and 207: check the whole sentence? Response 8: This sentence was checked and rephrased.
Comments 9: Line 294: The pots were not are, Line 295: the temperature was not is and humidity was not is …, Response 9: Agree. We have changed the tenses of Line294 and Line295 to the past tense
Comments 10: Line 379: investigation, Line 380: formal analysis and investigation, Line 381: funding, Line 382: investigation Response 10: Agree. We have rewritten the words marked on line 379-382 in the correct format as requested by the reviewer.
Comments 11: Line 454: position 18 and 28 this same Response 11: Agree. Thanks to the reviewer's reminder, 18 and 28 are indeed the same and we have revised them in the reference.
Comments 12: Line 494: empty space Response 12: Agree. We have made changes to the reference format located on Line494.
|

Reviewer 2 Report
Comments and Suggestions for Authors
Frankly, the manuscript designed and prepared well. This type of manuscript is so important for weed management, especially if flooded rice. Minor corrections are required
Introduction
Before talking about penoxsulam herbicide, write a general paragraph clarifying the advantages and disadvantages of herbicides use in weed control.
Lines 76-79, support the sentence "To achieve better weed control……………….. spraying herbicides" By the following citations:
https://doi.org/10.1016/j.cropro.2021.105755
https://doi.org/10.1007/s42729-020-00356-1
https://doi.org/10.1080/03650340.2013.866226
Methods
Well done. However, in pot experiment, the level of irrigation water layer should be mentioned.
Author Response
|
Response to Reviewer 2 Comments Thank you for your letter and for the reviewers’ comments concerning our manuscript entitled "Post-emergence water dispersal application provides equal herbicidal activity to Echinochloa crus-galli and rice safety as foliar spraying in penoxsulam". (ID: plants-2708058). Those comments are all valuable and very helpful for revising and improving our paper, as well as the important guiding significance to our researches. We have studied comments carefully and have made corrections which we hope meet with approval. Revised portion are marked in colored words or using the track changes mode in the paper. The main corrections in the paper and the responds to the reviewer’s comments are as flowing:
Responds to the reviewer’s comments:
|
|
Point-by-point response to Comments and Suggestions for Authors |
|
Comments 1: Introduction: Before talking about penoxsulam herbicide, write a general paragraph clarifying the advantages and disadvantages of herbicides use in weed control. |
|
Response 1: Thanks for your suggestion. We agree with the reviewer’s advice that the advantages and the disadvantages of herbicides have been added in this paper.
|
|
Comments 2: Lines 76-79, support the sentence "To achieve better weed control……………….. spraying herbicides" By the following citations: https://doi.org/10.1016/j.cropro.2021.105755 https://doi.org/10.1007/s42729-020-00356-1 https://doi.org/10.1080/03650340.2013.866226 |
|
Response 2: Thank you for pointing this out. We have added three references recommended by our reviewers to this article.
Comments 3: Well done. However, in pot experiment, the level of irrigation water layer should be mentioned. Response 3: Thanks for your suggestion. In the material method in line 302 of the paper, it has been mentioned that the depth of the water layer is 2cm |
|
|
